# The Post-Pandemic Transformation of Art and Architecture Libraries

**Rose Orcutt** [1,*] , **Lucy Campbell** [2] , **Maya Gervits** [3] and **Barbara Opar** [4]

1   University Libraries, University of Buffalo, Buffalo, NY 14214, USA
2   Richard P. Welsh Library, NewSchool of Architecture and Design, San Diego, CA 92101, USA
3   Barbara and Leonard Littman Architecture and Design Library, New Jersey Institute of Technology, Newark, NJ 07102, USA
4   King + King Architecture Library, Syracuse University, Syracuse, NY 13244, USA
*   Correspondence: rmorcutt@buffalo.edu

**Definition:** This entry paper considers the impact of the COVID-19 pandemic on the processes and functions of art and architecture libraries in North America and distinguishes between temporary changes and those that will endure and are here to stay. COVID-19 impacted all aspects of human life, placing tremendous stress on institutions and individuals globally. Academic libraries responded to the crisis by bringing resources to communities remotely and keeping constituents engaged to maintain a sense of normalcy. While libraries in schools of architecture, art, and design, responded similarly to other academic libraries, they also had unique needs. This entry paper is informed by two surveys of art and architecture library staff and faculty, alongside a preliminary literature review. The results of the first survey were published in Art Documentation and the results and analysis of the second survey are forthcoming. Both temporary and long-standing changes were implemented to ensure uninterrupted service in academic institutions. Temporary solutions included extending loan periods, quarantining materials, enforcing social distancing, and expanding document delivery. Changes that will endure post-pandemic include the increased acquisition of digital materials, remote instruction and reference consultations, increased resource access, and the utilization of a vast array of technologies.

**Keywords:** COVID-19 pandemic; post-pandemic; libraries; online access; remote services; digitization; transformation; art and architecture; academic libraries

## 1. History

The recently published report *The Library of the Future* indicates that "the library profession, much like the library itself, is undergoing profound change" [1]. Indeed, following the COVID-19 pandemic, many library activities have moved online. Librarians have adapted and become "vital partners with scholars, instructors, students, and patrons in a way few had experienced before" [1].

December 2019 saw the first outbreak of the coronavirus disease in Wuhan, China. By mid-March 2020, all 50 of the United States had reported cases, leading to a rapid shutdown of businesses, cultural and religious institutions, and public and private educational facilities, including libraries. Few if any were prepared for the changes necessitated by the shutdown. Preparedness manuals offered little relevant content, and practices were rarely uniform even within institutions. We have now reached a point in the operation of libraries where predictions can be made about temporary workarounds versus changes that are here to stay.

Beginning in 2021, publications focused on library responses to COVID-19 were released, providing information on both public and academic libraries [2–5]. The majority address particular case studies, or specific library functions such as research productivity,

management, copyright, or technological advancements. *The Handbook of Research on Library Response to the COVID-19 Pandemic* takes a scholarly approach to understanding how libraries transformed during the global health emergency. Chapters highlight a range of topics including crisis leadership, data management, and virtual programming. Library types from around the world are also represented. Chapters include *Opportunities and Challenges Offered by the Effects of the COVID-19 Pandemic on Academic Libraries* which proposes, "libraries cannot afford to close down their formal duties of providing information to the users . . . The libraries should be in continuous coordination with the researchers conducting research in the field of the cure of the COVID-19" [6]. However, little attention has been paid to the specific response of architecture, art, and design libraries.

This entry applies to the information science field and information professions, more specifically to those in the art, architecture, and design disciplines. This content is informed by an exploratory review of current literature addressing responses and changes to academic library functions resulting from the COVID-19 pandemic, alongside two extensive surveys of library staff and faculty users. These surveys were circulated during the early and late stages of the pandemic. Detailed findings have been published [7].

## 2. Evaluation

### 2.1. Customer Service and Reference

#### 2.1.1. Temporary Changes

Temporary measures allowed for the continuation of library services while addressing staff safety concerns. Collaboration between institutions was critical in formulating the format and extent of services and safety protocols. Organizations such as The Center for Disease Control and Prevention (CDC), and National Endowment for the Humanities (NEH), provided overall guidance, while, The International Federation of Library Associations and Institutions (IFLA), and American Library Association (ALA) shared library-specific best practices [8]. Many provided online discussion space and webinars, for example the Medical Library Association (MLA) published guidelines for appropriate COVID-19 preparation and practice [9]. Across architecture and design libraries, specific recommendations were shared via association and membership listservs. Measures included curbside pickup, social distancing signage, sanitizing spaces, alternative work schedules, restricting visitor numbers, the promotion of e-resources and e-services, and quarantining circulating materials (Table 1). In architecture, art, and design libraries, access to visual print materials remains integral to research practices. Quarantine and curbside pickup were dominant temporary responses. For example, 72% of architecture libraries reported offering curbside pickup services in Spring of 2021 [7]. However, these services have been terminated as patrons returned. After months of closures and restricted access, libraries reported 40–60% return of visitors in August 2022, compared to pre-pandemic levels, a number that is expected to increase [10].

**Table 1.** Circulation changes, temporary and/or permanent. These findings are informed by surveys and a review of the literature.

| Circulation Changes | Temporary | Permanent | Notes |
|---|:---:|:---:|:---:|
| Building closures and/or limited access | X | | |
| Reduced Operating Hours | X | | A few institutions permanently adjusted hours of operation |
| Curbside Pickup | X | | |
| Check out to lockers | | X | |

**Table 1.** *Cont.*

| Circulation Changes | Temporary | Permanent | Notes |
| --- | --- | --- | --- |
| Extended loan periods | X | | Some institutions may have studied and adjusted loan period structure post COVID-19 |
| Fine forgiveness | X | | |
| Glass/plexiglass desk partitions | | X | |
| Staffing working from home full or part time | | | A few institutions have continued to allow staff to work off site some of the time. This service however does not lend itself well to off-site activities |
| Free mailing items to patrons | X | | Mailing has continued but only for distance patrons- over 35 miles |

### 2.1.2. Here to Stay

Following the COVID-19 pandemic, the concept of the art and architecture library being "open" no longer necessarily means the doors are unlocked. Instead, staff and materials are accessible and responsive to patrons digitally. As summarized by Dr. Todaro, "You can't see us, but we are here to help you" [11]. For libraries that established robust remote service models, service hours and access to physical spaces are now two different and distinct concepts. This change in both how libraries perceive their services and what users expect is likely to remain long term.

Guidelines changed over time as new information became available, for example studies on the effects of contact with different types of paper [12]. However, the guidelines provided by associations and professional organizations for quarantining were inconsistent and often contradictory. The recommendations varied between 2 and 8 days, and in September 2021, Northeast Document Conservation Center (NEDCC) suggested determining the length of the quarantine period by assessing the level of risk in each community [13]. Professional organizations have begun to develop standards and plans for managing communications, public health crisis, and access during lock down situations [14]. It is anticipated that with the benefit of hindsight some will publish informed and extensive guides that will serve as useful future-proofing tools.

Reference services have been permanently changed by COVID-19. When physical access to collections became impossible, reference librarians continued to assist patrons in creative ways. Patrons were referred to items available online, e-resources were ordered as surrogates, and staff with partial access to facilities scanned items for users (Table 2). These services have now become standard expectations where they were once exceptions.

Architecture literature continues to be heavily print reliant. The major periodical index, the *Avery Index to Architectural Periodicals*, is a bibliographic citation index, and the only reliable database to comprehensively search image saturated architecture periodicals. This continues to be a unique characteristic of libraries in the discipline. This discovery issue is specific to the field and this resource. While some older content has been made freely available online, the number of such titles is limited. The Avery also indexes many journals for which there is no online content. *The Architectural Review* has provided consistent online access for less than ten years, with very selective access to some older content. *Architectural Design* online is available only from 2005. Older content is important in architecture for historical and precedent research. Architectural research is image based and the quality and clarity of online images is paramount. With limited aggregator tools which in older cases are not sufficiently precise or uniform to direct patrons to online content. For example, a reference to the Seagram Building restaurant is cited as *Architectural Record.* 1959: 201–204. While there is freely available digital access to this journal through the *US Modernist Library*, the user needs more information to locate the source. Reference work is multifaceted, and in the case of architecture often involves in-depth content knowledge, for example detailed

drawings of specific projects. It cannot be comprehensively conducted without access to materials, and older titles are often unavailable in e-formats. Architectural design is particularly challenging as few architectural monographs are digitized. While databases such as *DETAIL Inspiration* offer drawings, access is limited to newer projects. Similarly, architectural archives may be open access, but many offer low-resolution drawings. Others, such as *Le Corbusier Plans* are costly and extremely specialized.

**Table 2.** Reference Changes, temporary and/or permanent. These findings are informed by surveys and a review of the literature.

| Reference Changes | Temporary | Permanent | Notes |
|---|:---:|:---:|---|
| Lack of access to physical content | X | | |
| Greater reliance on purchased e-content | | X | Many libraries are now selecting e-preferred for approval plans and some other content |
| Greater reliance on open access e-content | | X | Archives such as *US Modernist* provide digitized content of core journals in the discipline |
| Lack of digital access to older content | | X | While this is evolving, there is no consistent or intentional approach to digitizing older monographs or journal titles. |
| Concierge approach to reference work | X | | Post COVID-19, many patrons have come to expect services like paging of books |
| Changes to how reference performed—reference interview curtailed or shortened | X | | During COVID-19, many librarians shortened this process and simply supplied the patron with the desired content |
| Expansion of chat services | | X | Patrons who had not previously used chat came to rely on this service |
| Zoom reference | | X | Hybrid staffing has made this a desirable option for staff and some patrons |
| Hybrid work schedules | | X | Most institutions have accepted hybridity as the new norm |

### 2.2. Outreach and Engagement

2.2.1. Temporary Changes

Remote attendance changed outreach activities. Digital newsletters, emails, blogs, online exhibitions, and social media existed before the pandemic. However, reliance on these tools to promote resources, programming, and mental health checks increased as librarians sought to connect with isolated faculty and students. Online research guides became a prominent tool to present resources for class instruction. One analysis by Springshare found librarians created 26,000 new research guides in response to the pandemic [15]. Zoom sessions, Google Hangouts, and Microsoft Teams promoted scheduled public events such as book talks, lecture series, and presentations to patrons and sometimes increased attendance in comparison to in-person events, as people looked for ways to connect. Although these practices will continue to feature in library services, patrons and staff desire more in-person connections. Face-to-face activities are once again increasing.

2.2.2. Here to Stay

Despite the growing desire to meet in person, these events continue to incorporate online components. The hybrid environment is here to stay and is already impacting

architectural education. A core practice in architecture and design education is the Lecture Series, in which prestigious guest speakers are invited to campus. Typically open to the public, these lectures are now being broadcast live online for free, opening them to a new audience of students and learners around the world. For libraries, the opportunity to support these efforts by providing resources and content has added value to their services. Libraries can promote lectures happening at universities in Hong Kong, Germany, or Brazil, while connecting users to relevant and unique content in their own collections.

*2.3. Collection Development*

2.3.1. Temporary Changes

Acquisition efforts shifted to electronic as opposed to analog materials, including books, journals, databases, and streaming videos. Some libraries implemented Controlled Digital Lending (CDL) to allow digital circulation of print materials including course reserve materials, and publishers bridged the gap by providing online access to critical resources for free or at reduced rates [16–18]. Initially the Internet Archive offered some 1.4 million titles through free CDL, although this initiative was quickly stymied by publisher concerns around intellectual property rights [19].

While collection development was pushed to a hybrid model, COVID-19 also led to budget concerns and policy changes. While academic libraries already preferenced e-formats in many disciplines, the arts retained greater flexibility and subject selectors made case-by-case judgments. To ensure 24/7 access, libraries mandated e-formats across disciplines, expanding access but drawing attention to the issues of e-format availability in the architecture and design disciplines.

2.3.2. Here to Stay

With decreased budgets, creative publishing methods, and where feasible the adoption of digital first strategies, librarians are cautiously participating in production processes for scholarly content. Open access publishing and Open Educational Resources (OER) are increasingly attractive alternatives as growing interest results in falling costs, expanded access, and most importantly, improved quality [20,21]. Although they have existed since the late nineties, OER have a reputation of low quality and inaccuracy. COVID-19 has changed those attitudes and signaled a willingness amongst educators to adopt and grow these free resources, making them available to remix, reuse, and adapt to course objectives.

While cost saving methods such as demand-driven acquisition, resource sharing, and consortiums are the new normal, print based collections in art and architecture will continue to struggle to maintain contemporary literature holdings. COVID-19 induced gaps in collections are unlikely to be filled because of because print budget has been reduced. Emphasis on lounge and study spaces and collaborative services with other departments has also led a trend towards reevaluating print holdings [1].

*2.4. Digital Resources*

2.4.1. Temporary Changes

Electronic books have existed since the digitization of the Declaration of Independence in 1971. However, widespread acceptance of the format really began just twenty years ago, and until recently the humanities have been particularly resistant [22–24]. Although they provide perpetual access, eBooks present issues such as managing logins, simultaneous use models, varying vendor platforms, and print quotas. eBook packages are rented and susceptible to continuous change. According to publisher agreements, content may disappear suddenly, creating uncertainty. Many libraries today employ evidence-based selection methods, so key titles are subject to removal if not consulted regularly. Despite these known issues architecture libraries were temporarily pushed to purchase eBooks exclusively.

According to a Motion Picture Association report, the number of subscribers to streaming services during the pandemic reached 1.1 billion [25]. The increased consumption of streaming media and vendor pricing models created a challenge for libraries to meet

patrons' needs, especially in architecture libraries where film is an important visual learning resource. Some libraries developed policies and practices for converting DVDs to streaming content [26,27]. However, no standard controlled lending policies have been established to date. Institutions were slow to produce such copies, and a murky copyright environment meant most were only comfortable using this practice for required course materials or limited film snippets. With the return to in-person teaching, these efforts were largely abandoned.

### 2.4.2. Here to Stay

With the assumption that eBooks are favored by many patrons, some states such as Maryland and Rhode Island have introduced legislation mandating that eBook publishers must allow libraries to purchase eBooks at a "reasonable price" ensuring fair and balanced lending. These bills can be preempted by the federally mandated Copyright Act, as happened in New York. Nevertheless, this highlights the need for legislation and the possibility of bringing the issue to the state level to preserve the library's mission to make books available to the public at a reasonable cost [28].

Without a doubt, streaming media is both here to stay and has many advantages over outdated formats. Despite the TEACH Act, online teaching made it more problematic to use DVDs and resulted in increased use of platforms such as Kanopy, Swank, Artfilms Digital, Academic Video Online, OnArchitecture, and Pidgeon Digital. While these vendors provide robust collections, they are far from exhaustive. As a result, libraries must regularly negotiate streaming rights for specific titles which, although costly and available for a predetermined time, have become standard practice [29].

Libraries are continuing to struggle with providing online access to both new and older titles. While many vendors such as JSTOR provided expanded temporary or free access to certain collections during COVID-19, the industry has not responded satisfactorily to the ongoing needs of libraries [30]. Rather than provide unlimited or DRM free access to titles, many vendors have chosen to return to 1 or 3 simultaneous users. Multiple vendors provide access to popular titles, but few address the need for key older titles, especially in architecture. eBooks costs are rarely consistent, and while a few database vendors such as MADCAD are willing to expand access to certain resources for specific periods of time, this practice is rare. Surveying libraries and vendors for input on resource needs could be mutually beneficial.

### 2.5. Technology

#### 2.5.1. Temporary Changes

Technology empowered libraries to maintain continuity and opened additional avenues for cooperation. Through collaborative efforts, they offered settings for new interactions and fostered engagement. Technology played a critical role in enabling online teaching, communications, and outreach. Popular communication software included Zoom, Jabber, Webex, and Google Meet. Mentimeter, Top Hat, and Canvas Tool Box helped increase interactivity, and smart video conferencing equipment such as Owl Pro facilitated both online teaching and group meetings. In architecture education, collaborative design software such as the Miro online whiteboard enabled studio culture to persist in a digital world. However, many of these solutions have since been discarded or neglected in favor of traditional studio practices.

#### 2.5.2. Here to Stay

The high level of flexibility which technology facilitated will remain critical in a post-COVID-19 world. Some specialists predict that hybrid and HyFlex offerings will likely serve as a baseline for future operations [31,32]. Libraries and librarian roles are changing, and staffing patterns will follow suit. While many libraries are now permitting flexible scheduling and work from home, there are no uniform policies. Staffing patterns and roles and responsibilities vary from institution to institution. Guidelines and best practices

could be established through a consensus of professional organizations. The move to hybrid delivery both in the classroom and for student services has increased expectations of technological expertise of staff. Remote communication techniques have been established and will continue.

Increased collaboration and aggressive adoption of new technologies became typical in education and research during the pandemic. Asynchronous learning provided opportunities for a more flexible educational process supporting students who were not always in the same time zones or unable to attend synchronous lectures. It allowed for increased participation and enhanced accessibility. At the same time, virtual meetings offered more dynamic and collaborative engagement, while requiring more nuanced scheduling, technical competence, and adequate infrastructure.

*2.6. Remote Work Arrangements*

2.6.1. Temporary Changes

The pandemic required almost everyone to work remotely. While this created flexibility in schedules, access to key resources were hard to obtain as the majority of architecture core texts remain print based. Remote work created blurred boundaries between work and home life. Librarians who cared for family, children, or the elderly had to balance work schedules with these responsibilities. The influx of virtual services such as Zoom, Google Meets, Slack, chat services, and similar platforms was a considerable positive aspect that facilitated interactions with students, staff, and faculty. These services allowed librarians to continue and even strengthen communication, teaching, and reference skills, proving the profession could function remotely [33].

2.6.2. Here to Stay

With the realization that work could be done at home, employees challenged the preceding notion of incapability to telecommute and pushed for more flex time. Librarians are re-assessing their work schedules; keeping the efficient productive responsibilities and obligations while focusing on efficient, uninterrupted operations. Working on-site may be necessitated by in-person meetings, class instruction, appointments, and faculty and student interactions. Virtual meetings, classes, appointments, and other activities could be accomplished remotely without the need to commute or be interrupted. Research has found that hybrid and flex schedules increase production and efficiency [34]. The ability to work remotely can reduce stress and help with individual mental health and wellbeing [35]. However, job related stresses such as inadequate pay, increased duties with no extra compensation, and unhealthy schedules has also led to the phenomenon of 'quiet quitting;' a push back on being undervalued [36]. Realistic goals should be set by supervisors who understand the employees' need for flexibility, thus creating a supportive workplace culture based on communication and mutual support [37]. The post-pandemic art and architecture library must seek new ways to think about information services and patron connections. Libraries must address change and acknowledge that the situation is different now and seek alternative ways of measuring the impact the library has on the campus community other than foot traffic and in-person interactions. In the aftermath of COVID19, additional assessment should be made to determine what has changed, what has improved and what issues still need resolution.

COVID-19's impact on academic art and architecture libraries has been transformative and clearly challenged librarians to creatively adapt. However, lessons learned have enabled libraries to be nimble, adapt quickly, learn new ways of doing things and expand services, practices, and goals.

**Funding:** This research received no external funding.

**Institutional Review Board Statement:** Not applicable.

**Informed Consent Statement:** Not applicable.

**Conflicts of Interest:** The authors declare no conflict of interest.

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
