# Peer review of "The Post-Pandemic Transformation of Art and Architecture Libraries"

_encyclopedia, doi:10.3390/encyclopedia2040131_

Round 1

Reviewer 1 Report

Comments on The Post-Pandemic Transformation of Art and Architecture Libraries

Definition. I would not say any changes in libraries are “irreversible.” That’s not the same as “here to stay.”

The abstract discusses a survey but there is no further discussion of it. It’s unclear if the statements of opinion in the piece are a result of the survey or the authors’ own opinions (when there are no citations). It’s unclear if Tables 1 and 2 reflect survey findings or something else (it’s hard to believe that summary findings showed that “mailing has continued but only for distance patrons over 35 miles,” or plexiglass partitions will persist, for example).

3.1 should cite the OCLC REALM project. 3.1.1 “quarantine of circulating materials” is listed twice.

3.1.2. First paragraph should clarify this applies to A&A libraries, because academic libraries have been “open” virtually for at least 25 years, longer if you count remote access to online catalogs.

Second para: “Guidelines…were inconsistent and contradictory” – should be a citation for that.

“Professional organizations have yet to develop future standards for access” is unclear – does that mean access to spaces? Access to physical materials?

The para on Avery Index is unclear how it relates to physical collection access. An incomplete citation sounds like a discovery problem even in a physical library.

3.3 and 3.4 are basically the same topic and could be combined.

3.3.1 does not address course materials but 3.3.2 does. “Digital first” collecting is not related to OER, yet OER is introduced this way.

“print-based collections in art and architecture will continue to struggle to maintain contemporary literature holdings.” Is this because of digital-first? Or Space? Is it at all related to COVID?

3.4.1 First paragraph is about problem with ebooks. Where are the COVID-related temporary changes?

3.4.2 “With the certainty that eBooks are favored by patrons….” Patrons of A&A libraries? 3.4.1 pointed out how humanists have been resistant, and 3.1 and 3.3 pointed out how many A&A materials are print only. This section about public-library-focused legislation on ebook pricing and vendor use restrictions doesn’t seem relevant to a discussion of post-COVID changes. Much more relevant would be legislation related to Controlled Digital Lending, which would affect how print-centric collections like A&A could provide remote collection access in the future.

“Circulating hardware… will remain in place” requires some data, or citation. Many libraries have already cut back on these services, or found them of *less* use to patrons post-COVID (though I also do not have citations, just anecdotal information).

3.6 might be better characterized as “remote work” or “hybrid work locations.” Work/life balance is one factor in such arrangements, but it does not characterize the change.

“Librarians are reassessing their work schedules; keeping the efficient productive responsibilities and obligations while eliminating repetitive ineffective operations.” The 2nd clause does not related to work schedules. The implication is that on-site work is inherently repetitive or ineffective?

“While virtual meetings….” Is a sentence fragment.

“However, job-related stresses….” – are the authors proposing this is “here to stay”?

“Libraries must …. Seek alternative ways of measuring the impact the library has on the campus community.” Is this related to post-COVID because impact was previously based on foot traffic? Or in-person interactions?

“Covid’s impact on academic art and architecture libraries has been transformative and clearly stretched librarians’ abilities.” That is quite negative, and does not reflect the tenor of the piece. I would suggest “clearly challenged librarians to creatively adapt.”

Author Response

On behalf of my coauthors, I would like to thank you for the opportunity to revise our entry The Post-Pandemic Transformation of Art and Architecture Libraries. We found the comments and recommendations thoughtful and helpful in improving the manuscript. We have accepted and incorporated all the suggestions except for combining 3.3 Collection Development and 3.4 Digital Resources. We felt the content was distinct enough to keep them as separate topics.

We have tracked the changes on our revised manuscript. Please see the attachment.

Reviewer 2 Report

The paper is of interest as talks about the changes experienced for a type of libraries - arts and architecture libraries- in some universities during COVID 19 and the temporary and permanent changes brought by it. The objectives are clear and the results well organized and consistent.

Congratulations.

Author Response

On behalf of my coauthors, I would like to thank you for your comments on our entry The Post-Pandemic Transformation of Art and Architecture Libraries

Reviewer 3 Report

Dear authors,

- please add more relevant references,

- correct formal site ( page alignment),

- add section (methods)

- improve conclusions and future research

Thank you.

Author Response

On behalf of my coauthors, I would like to thank you for the opportunity to revise our entry The Post-Pandemic Transformation of Art and Architecture Libraries. We found the comments and recommendations thoughtful and helpful in improving the manuscript. 

We added four more citations. The page alignment is dictated by the provided template and therefore difficult to change but we did the best we could. The Method section was not part of the template but we did address it better in the Abstract. A conclusion was also not part of the template but we reworded the text to make it clearer.

Reviewer 4 Report

Very well written, interesting, and insightful. A few minor edits recommended...

Covid, COVID, COVID-19, and COVID19 are all used---could these be reviewed and one acronym (e.g., COVID-19) be used throughout, please

Table 1 and Table 2, the column heading for Temporary/Permanent is confusing...clearer would be to have two column heads: one that reads Temporary (followed by Xs in that column) and then one that reads Permanent (followed by Xs in that column.

Author Response

On behalf of my coauthors, I would like to thank you for the opportunity to revise our entry The Post-Pandemic Transformation of Art and Architecture Libraries. We found the comments and recommendations thoughtful and helpful in improving the manuscript. 

We made all the COVID-19 language uniform and reworked the columns in the tables.

Round 2

Reviewer 1 Report

The manuscript responded to my comments. It is improved.

Reviewer 4 Report

Terrific article. The updates make it a very strong, clear, and meaningful entry for the Encyclopedia.